# Numerical Investigation by Cut-Cell Approach for Turbulent Flow through an Expanded Wall Channel

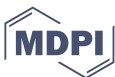

Ramzy M. Abumandour [1], Adel M. El-Reafay [2], Khaled M. Salem [2,*] and Ahmed S. Dawood [1]

1 Basic Engineering Sciences Department, Faculty of Engineering, Menoufia University, Shebin El-Kom 32521, Egypt
2 Department of Basic and Applied Science Engineering, Arab Academy for Science, Technology and Maritime Transport, Smart Village Campus, Smart Village 32521, Egypt
* Correspondence: khaled.selem1997@adj.aast.edu

**Abstract:** The expanded wall channel backward-facing step (BFS) and axisymmetric diffuser plays an important role in the society of fluid dynamics. Using a cut-cell technique is an established new method to treat the inclined wall of an axisymmetric diffuser. Cut-cell handle to reach the shape of the inclined wall, an axisymmetric diffuser and complex geometry. It helps treat the boundary condition at the wall in an accurate physical way. The turbulent flow through the geometries is solved by using Reynolds averaged Navier-Stokes equations (RANS) with the standard k-$\varepsilon$ model. A self-built FOTRAN code based on the finite volume method with the Semi-Implicit Method for Pressure Linked Equations (SIMPLE) algorithm for pressure velocity coupling is established and examined with published experimental data for two different geometries backward-facing step (BFS) and axisymmetric diffuser. The results of the new technique reflect good agreement between the numerical results and the experimental data. A parametric study of the impact of area ratios (2, 2.5, 3, 3.5) in a backward-facing step on pressure, velocity, and turbulent kinetic energy. The angles ($7°$, $10°$, $14°$) and area ratios (2, 2.5, 3, 3.5) effect of an axisymmetric diffuser on the streamlines, local skin friction, pressure, velocity, turbulent kinetic energy, and separation zone.

**Keywords:** axisymmetric diffuser; backward-facing step; turbulent flow; numerical method; finite volume; cut cell; RANS; CFD





## 1. Introduction

Diffusers and backward-facing step (BFS) are the most important flow devices in the industry, especially in hydraulic machines, combustors and flow over airfoils, among the many applications of adverse pressure gradient flows. Consequently, research into turbulent flow through diffusers and backward-facing step (BFS) has been a crucial area of interest for fluid mechanics specialists, so engineers frequently uses in their work. They desired to raise a flow's static pressure by lowering its velocity, though frequently with significant losses and they were typically simple in design. At the expanding section in the diffuser, a separation bubble forms and quickly spreads across one of the diffuser's walls and in a down-stream pipe where additional pressure recovery takes place, the flow eventually reattaches [1,2]. According to clear diffusers and backward-facing step (BFS) shapes and the availability of a high-quality velocity measurement, this diffuser and backward-facing step (BFS) have become a popular test case for detecting moderate separation, validating numerical models and this flow is regarded as a crucial test that can be used to compare various turbulence models [3]. To simulate this flow in different geometries, a number of turbulence models have been used, including Reynolds-averaged Navier-Stokes (RANS), Large eddy simulation (LES), and Direct numerical simulation (DNS) [4–7].

When the fluid enters an expansion, the static pressure increases at the expense of the kinetic energy flow, which usually decreases with increasing pressure. The core spreads out behind the expansion and forms a surface of separation to cut itself off from the remaining

fluid. As the surface of separation reaches moderate to high Reynolds numbers (Re), it becomes unstable and creates turbulent eddies in a recirculation or free-mixing stall zone. The eddies develop and eventually disappear, recirculation with separation results in increased turbulence, high pressure loss, a faster rate of mass and heat transfer [8,9]. This mainly occurs because the fluid flows against an opposite pressure gradient, which forces the fluid molecules to follow an opposite path close the wall of the larger tube just before the step rise. After a while due to the expansion present, streaming can reach the desired speed and reach a fully developed [10–12].

Most mathematical problems in this field are very challenging, and generally there is no straightforward solution. The state of a flow at any given place in space-time is defined by a set of partial differential balancing equations, which regulate fluid dynamics simulations. It is complex and maybe even impossible, to find the exact solution. Due to the tremendous advancements in computational technology, numerical solution has grown in popularity and is now a crucial skill for scientists and engineers [13]. There are many numerical techniques such as (Finite Element Method (FEM) [7,11], Finite Volume Method (FVM) [2,4,9,14–17] and Finite Difference Method (FDM) [18–21]). Solving this system with these partial differential balance equations by convert the system of partial differential equations into a system of algebraic equations to get the results, and all of these techniques have different ways to solve the problem [22,23].

Many authors were interested with the adverse pressure gradient devices. An axisymmetric diffuser has been studied by many authors as [11,12,24,25]. A asymmetric diffusers [13,15,16] and backward facing step (BFS) [9,14,18,19] studied by many authors to predict the turbulence models.

Lee et al. [12] studied an axisymmetric diffuser with three opening angles of 2°, 4°, and 8°, by applying a direct numerical simulation (DNS). There was work conducted to examine the turbulence statistics and coherence structures in a fluid flow and compare the numerical results with the experimental results of Singh and Azad [26]. They treated the gradient wall in this work by applying the grid transformation with polar coordinates. They exhibited the axial velocity, radial velocity and pressure recovery. When the opening angle increased, the maximum Reynolds shear stress increased. EL-Askary et al. [24] treated the axisymmetric diffuser wall by assuming the wall as a ladder step. Four different axisymmetric diffuser geometries were tested at various Reynolds numbers and mass loading ratios. The investigated Chen-Kim turbulence model illustrated the axial velocity, radial velocity, pressure recovery, local skin friction, and streamlines. This result indicated how the Reynolds number affected the axisymmetric diffuser. The experiment [25] was appeared the different Reynolds numbers and angles in the wide-angle diffuser. This experiment discovered how the Reynolds number and angles affected velocity and pressure recovery. With a different inlet condition, this experiment demonstrated a big difference when the velocity increased from 10 m/s to 20 m/s; the wall static pressure recovery increased to 8.31%. Increasing the Reynolds number helped to increase pressure recovery.

El-Behery et al. [15] discussed the asymmetric Diffuser by using Reynolds average Navier-Stokes equation (RANS) with the software ANSYS Fluent and a finite volume approach with respect to different turbulence models, which were called standard k-$\varepsilon$, low-Re k-$\varepsilon$, standard k-$\omega$, SST k-$\omega$, v2-f, and RSM models, were available as standard features in the code. However, it displayed the axial velocity, turbulent kinetic energy, pressure recovery, and local skin friction. The findings demonstrated that when an adverse pressure gradient was present, the standard k-$\omega$, SST k-$\omega$, and v2-f models performed much better than other models. Singh and Mukhopadhyay [16], was tested an asymmetric diffuser with RANS and a finite volume approach by commercial fluent code. They compared a several turbulence models known as "low Re k-$\varepsilon$", SST k-$\omega$, v2f and a variant of the Reynolds stress model (RSM). The outcomes were contrasted with the precise pressure and velocity data. In terms of pressure recovery and flow field prediction in the asymmetric diffuser, the Reynolds stress model (RSM) model performs well.

Thangam and Speziale [9]. Showed the impact of the difference between two turbulence models (the standard K-$\varepsilon$ model and the RNG K-$\varepsilon$ model) and the effect on velocity, eddies, streamlines, and turbulent intensity. This certainly did indicate that properly calibrated two-equation turbulence models in backward-facing step (BFS). Lu and Zhao [14] concentrated on backward-facing steps (BFS), forward-facing steps (FFS), and smooth ducts. Reynolds-averaged Navier-Stokes equations (RANS) for fluid flow and the Reynolds stress model (RSM) for turbulence were investigated using the software ANSYS FLUENT. The mean air velocity in various positions, flow drag, the flow velocity field, and the streamline were all investigated in three cases. They illustrated the distinction between backward-facing step (BFS), forward facing step (FFS), and the smooth duct.

On the other hand, a grid can be a structured or unstructured grid. The mesh of the structured grids is thought to be ordered, and the grid's specifics may be understood and communicated by using indicators, such as (I, J, and K), to identify the components and identify their neighbor's. In an unstructured grid, the intricacies of the grid are more complicated and challenging to obtain [27], and special lists must be kept to identify neighboring elements. When compared to structured grids, unstructured grids need more information to be recorded and retrieved, and changing element types and sizes might lead to greater numerical approximation errors [28]. The grid study of the diffuser is very important to know how to deal with the inclined wall; this means having a special case for this geometry and selecting the best way to choose the suitable solution [29].

There were many grid studies for diffuser such as, unstructured grid [30], grid transformation [12] and ladder step grid [24]. The unstructured grid, where each grid is dealt with according to shape. This method is considered one of the more difficult ones in the code process because studying each grid according to shape is difficult. Grid transformation is used to transform the physical inclined domain to a computational rectangular domain. But it should to transform all the differential equations for the movement of the fluid into the general form to be worked in this way. And ladder step grid [24], which treat the inclined wall as a summation of consequent of a sudden expansion at each grid, like a ladder. This method is considered the easiest, and scientists began to study it because it simple and easy to code it.

The Cartesian grid method has grown in popularity as a popular alternative to boundary-fitted methods for solving fluid flow in a fixed Cartesian grid. The domain must be partitioned into a grid that is compatible with the numerical methods used to solve the governing flow equations as part of a CFD analysis. For handling intricate geometry with several slopes, and curved surfaces, curvilinear or unstructured grids. Tetrahedral elements are currently the most popular solutions [31], to adequately define some regions, it might be necessary to use several connected curvilinear grids [32]. Hence, there is a cut cell technique [33], this technique is considered one of the existing techniques to reach the suitable results. It has been indicated that cell cutting helps reach a practical geometric shape. A Cartesian cut cell mesh can be generated by cutting the near wall cells to a new cell with a boundary fitted cell. The Cartesian cut-cell approach enforces precise conservation of mass, momentum, and energy even near the immersed border at a discretized level.

Tucker and Pan [33], they used the cut cells on a skewed channel, wall driven flow in an inclined box and flow over a cylinder. Among others [34–36], they used the cut cell for fixed solid boundaries, 2D inviscid and viscous fluid flow as well as 3D inviscid, viscous, and turbulent flows circular cylinder, non-inclined square cylinder and inclined square cylinder.

Moving body issues and free-surface/two-phase flows can both be solved using the Cartesian cut-cell method. For multiphase flows with moving bodies as well as fluid–structure interaction, the Cartesian cut-cell methods have also been developed. Describes the development of a hybrid cut-cell and ghost-cell technique [35]. In addition to the finite element and finite difference methods, the cut-cell method is also applicable. The coupling between velocity and pressure is a major concern for incompressible Navier-Stokes equations, and typically a staggered grid is used in most discretization methods, this

references describe the floating bodies, moving boundary problem, shaped solid boundaries and moving boundaries on a fixed Cartesian grid [36–41].

The most important application is the diffuser, for which we mentioned above that there are many methods, that helps reach the closest and most accurate results. Hence, for the best of available information reached, there are no published data use a cut cell technique to solve the turbulent flow through an axisymmetric diffuser. However, using the finite-volume method, the governing equations are discretized with specific attention paid to the handling of boundary cut cells, some cut cell types on a sloping surface, and using a hybrid cut cell/interpolation strategy, these cells are handled. The Cartesian cut-cell approach contains a number of advantages include in [33]:

- Grid creation is straightforward and easy to automate.
- Many high precision Integration diagrams assume a basic shape on a uniform Cartesian grid and are straightforward to construct.
- An adaptive grid optimization approach may be simply implemented on a Cartesian grid to give very high flow feature accuracy.
- By using a different mesh method, it is possible to avoid mesh flaws such extremely deformable cells that sometimes appear.

Our currently developed technique, which uses a cut-cell technique, is an established new method to handle the inclined wall of an axisymmetric diffuser. The near wall cells are treated as 5 faces for the new grid, one of them is the inclined wall. This helps treat the boundary condition at the wall in an accurate physical way. The wall shear stress was inclined and affected by the two equations (u, v) of momentum. Cut-cell helps to reach the shape of the inclined wall an axisymmetric diffuser and complex geometry. In house developed code was used to solve the backward-facing step (BFS) and axisymmetric diffuser using a cut-cell technique.

The aim of this study is to clarify the mechanisms governing of turbulent flow in the axisymmetric diffuser using a cut cell technique and backward-facing step (BFS). The numerical code has been confirmed by contrasting various experimental findings. The area ratios and the diffuser angles will be the two key variables in this investigation. The Reynolds average Navier-Stokes equations (RANS) with a closure turbulence model, known as the standard k-$\varepsilon$ model, were used to reduce the complexity of the turbulent flow. Studying the effect of different area ratios (AR) on the flow's velocity, streamlines, and separation bubbles behind the step wall. Figuring out whether the new method of cutting the cell may have an impact on the numerical code by searching for the angles difference that impacts the diffuser.

## 2. Mathematical Model

### 2.1. Problem Definition

The schematic diagrams of the two-dimensional backward-facing step (BFS) and axisymmetric diffuser are show in Figure 1. The angle is the variable, and the more diffuser is made when the angle goes up. The angle is 90 degrees and has two walls; it is a backward-facing step (BFS).

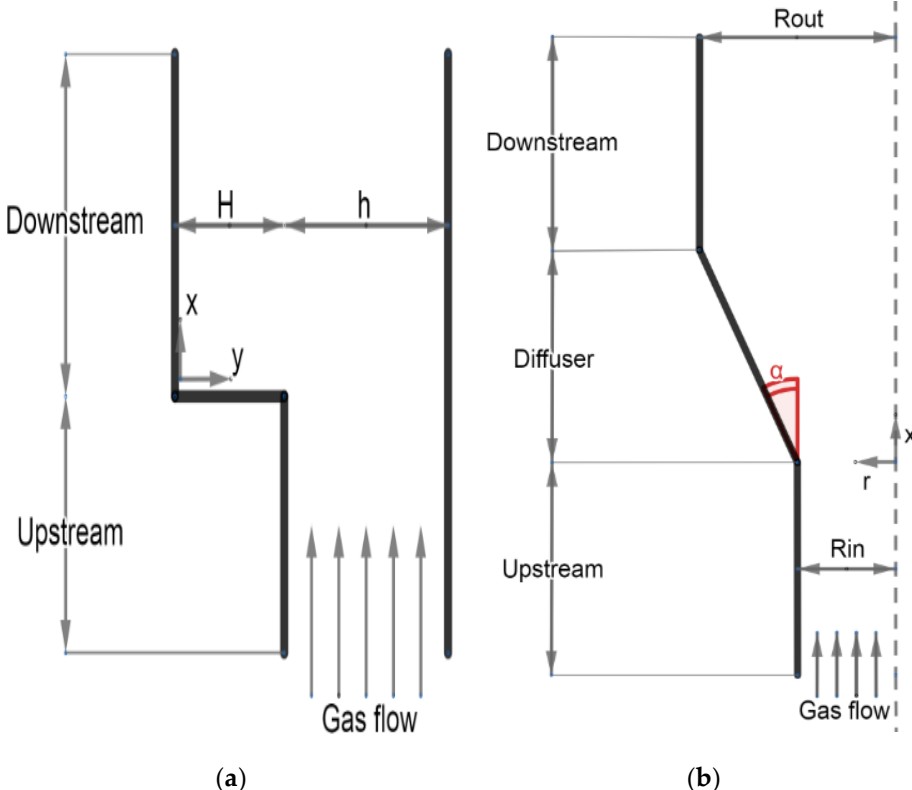

**Figure 1.** (**a**) Backward-facing step (BFS) flow geometry, (**b**) Sketch of axisymmetric diffuser step flow geometry.

### 2.2. Governing Equation

The incompressible flow through the axisymmetric diffuser and the backward-facing step have both been taken into consideration in this study. The Eulerian approach is used to code the numerical model. By solving Reynolds average Navier-Stokes equations (RANS) and the two-equation k-$\varepsilon$ turbulence model. However, the flow simulated by these equations in axisymmetric diffuser $(x, r)$, for backward-facing step (BFS) changes the axis to $(x, y)$ due to the two walls in duct.

Continuity equation:

$$\frac{\partial}{\partial x}(\rho U) + \frac{1}{r}\frac{\partial}{\partial r}(\rho r V) = 0.0 \tag{1}$$

Axial momentum:

$$\frac{1}{r}\left[\frac{\partial}{\partial x}(\rho r UU) + \frac{\partial}{\partial r}(\rho r VU) - \frac{\partial}{\partial x}\left(r\mu_{eff}\frac{\partial U}{\partial x}\right) - \frac{\partial}{\partial r}\left(r\mu_{eff}\frac{\partial U}{\partial r}\right)\right] = -\frac{\partial p}{\partial x} + \frac{\partial}{\partial x}\left(\mu_{eff}\frac{\partial U}{\partial x}\right) + \frac{1}{r}\frac{\partial}{\partial r}\left(r\mu_{eff}\frac{\partial V}{\partial x}\right) \tag{2}$$

Radial momentum:

$$\frac{1}{r}\left[\frac{\partial}{\partial x}(\rho r UV) + \frac{\partial}{\partial r}(\rho r VV) - \frac{\partial}{\partial x}\left(r\mu_{eff}\frac{\partial V}{\partial x}\right) - \frac{\partial}{\partial r}\left(r\mu_{eff}\frac{\partial V}{\partial r}\right)\right] = -\frac{\partial p}{\partial r} + \frac{\partial}{\partial x}\left(\mu_{eff}\frac{\partial U}{\partial r}\right) +$$
$$\frac{1}{r}\frac{\partial}{\partial r}\left(r\mu_{eff}\frac{\partial V}{\partial r}\right) - 2\mu_{eff}\frac{V}{r^2} \tag{3}$$

Turbulence kinetic energy:

$$\frac{1}{r}\left[\frac{\partial}{\partial x}(\rho r Uk) + \frac{\partial}{\partial r}(\rho r Vk) - \frac{\partial}{\partial x}\left(\frac{r\mu_{eff}}{\sigma_k}\frac{\partial k}{\partial x}\right) - \frac{\partial}{\partial r}\left(\frac{r\mu_{eff}}{\sigma_k}\frac{\partial k}{\partial r}\right)\right] = (G_k - C_D\rho\varepsilon) \tag{4}$$

Turbulence dissipation rate:

$$\frac{1}{r}\left[\frac{\partial}{\partial x}(\rho r U\varepsilon) + \frac{\partial}{\partial r}(\rho r V\varepsilon) - \frac{\partial}{\partial x}\left(\frac{r\mu_{eff}}{\sigma_\varepsilon}\frac{\partial\varepsilon}{\partial x}\right) - \frac{\partial}{\partial r}\left(\frac{r\mu_{eff}}{\sigma_\varepsilon}\frac{\partial\varepsilon}{\partial r}\right)\right] = \frac{\varepsilon}{k}\left(c_{\varepsilon1}G_k - c_{\varepsilon2}\rho\varepsilon + \frac{c_{\varepsilon3}G_k^2}{\rho\varepsilon}\right) \tag{5}$$

where the constant show in Table 1, and the generation term $G_k$ and effective viscosity $\mu_{eff}$ are expressed as follows

$$G_k = \mu_t\left[2\left(\left(\frac{\partial U}{\partial x}\right)^2 + \left(\frac{\partial V}{\partial r}\right)^2 + \left(\frac{V}{r}\right)^2\right) + \left(\frac{\partial U}{\partial r} + \frac{\partial V}{\partial x}\right)^2\right] \tag{6}$$

$$\mu_{eff} = \mu + \mu_t \tag{7}$$

where $\mu_t$ is the turbulent viscosity and can be calculated for the k-$\varepsilon$ turbulence models as:

$$\mu_t = \rho C_\mu \frac{k^2}{\varepsilon} \tag{8}$$

**Table 1.** Constants for k-$\varepsilon$ turbulence models used in the present study [13].

| Turbulence Model | $C_{\varepsilon1}$ | $C_{\varepsilon2}$ | $C_{\varepsilon3}$ | $C_D$ | $C_\mu$ | $\sigma_k$ | $\sigma_\varepsilon$ |
|---|---|---|---|---|---|---|---|
| Standard k-epsilon | 1.44 | 1.92 | 0.0 | 1.0 | 0.09 | 1.0 | 1.3 |

Where the $U$, $V$ are the velocity in 2d, $\rho$ is a density of fluid, $\mu$ is the laminar viscosity of fluid, $\mu_t$ is a turbulent viscosity.

### 2.3. Boundary Conditions

For the inlet condition, the turbulent kinetic energy and dissipation rate are considered to be constant, and the flow velocity is assumed to be a constant ($U_{in}(y) = c$) or a 1/7th power law profile equation [9] in x-direction and $V_{in} = 0$ in y-direction. With wall-function approximation, the no slip boundary requirements are assumed for solid walls [42].

$$U_{in}(y) = U_C * \left(1 - \frac{r}{R}\right)^{\frac{1}{7}} \tag{9}$$

where $U_C$ center line velocity.

The presence of flowing fluid past a stationary solid surface causes the fluid elements to deform under the action of shear forces. These shear forces make the velocity profile depends on the distance from the wall and the nature of the flow. A dimensionless quantity $y^+$ is calculated to determine the case where the near-wall flow as in Versteeg and Malalasekera [43] as follows;

$$y^+ = \frac{\rho C_\mu^{1/4} k_p^{1/2} \Delta y_p}{\mu} \tag{10}$$

### 2.4. Solution Procedure

The partial differential equations will be solved by estimating the finite volume with staggered grid using a hybrid approach and by using the SIMPLE algorithm to coupling pressure and velocity [43]. When the highest residual sum for all elements in the U, V and pressure correction equations is less than 0.1%, convergence is thought to have occurred. Here, using a new built numerical code based on a new technique of a cut cell method. Equations are solved on the FORTRAN program. The calculation of finite volume equations, Line by line, the TDMA (Three-Diagonal Matrix Algorithm) solution was used.

Convicted terms used a hybrid technique between the upwind differencing scheme with first order accuracy and the central differencing scheme with second order accuracy, while the diffusion terms used the second order accuracy of the central differencing scheme. These methods generally provided adequate accuracy, stability, and convergence. The

SIMPLE algorithm described by [43] was used for pressure–velocity coupling. The discretized equations are solved implicitly in sequence, starting with the pressure equation, followed by the momentum equations, by the pressure -correction equation, and finally by the equations for the scalars (turbulence variables). The convergence criterion consisted of monitoring (u,v and mass flow rate) values and variations of velocity profiles with iteration, reduction of several orders of magnitude in the residual errors.

The cells near the inclined wall have uniform shapes, so there are different five faces. This only applies to the u and v cells; the east and south cell faces require no modifications; the north and west cell faces require more modification and interpolation to get properties at cell face center(n',w') as in Figure 2, and we add a new face called the incline wall. Then all the equation will change as in [33]. The scalar cells (p, k, ε) don't need modifications.

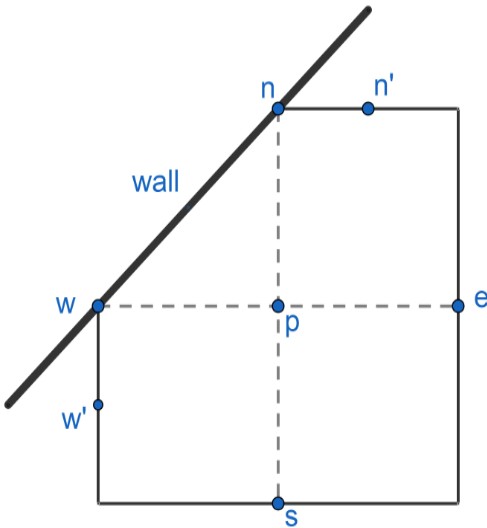

**Figure 2.** For an incline wall, A cut cell of an axisymmetric diffuser is used in the novel grid approach.

## 3. Results and Discussions

### 3.1. Validation of Code

The code is validated with different experimental results; Lu and Zhao [14], K.F. Yu et al. [18] and Bing W et al. [19] study the backward-facing step (BFS) through duct. Also, the experimental work of Singh and Azad [26] and Triesch and Bohnet [44] was used to validate the axisymmetric diffuser code.

All validation had been checked for the grid independent study, which used different grid sizes, non-uniform grids, and different inlet velocity conditions (uniform and 1/7th power law).

### 3.1.1. Validation of Backward-Facing Step (BFS)

Through comparisons with the experimental findings of [14] utilizing various grid resolutions ($300 \times 50$, $400 \times 100$, and $802 \times 150$) in stream-wise and normal to the wall directions respectively to choose the suitable resolution. The instance being examined contains the geometry of a rapid expansion channel with the upstream channel width ($h = 0.012$ m), step height ($H = 0.008$ m), and length of the down-stream ($L = 0.2$ m), the inlet gas velocity 5.5 m/s. Various stream-wise positions after the step wall of sudden expansion are plotted in Figures 2–4 with $X/H$ values of 2, 5, 7, 9 and 14, respectively. $H$ is the step height, while $X$ is the location measured from the step.

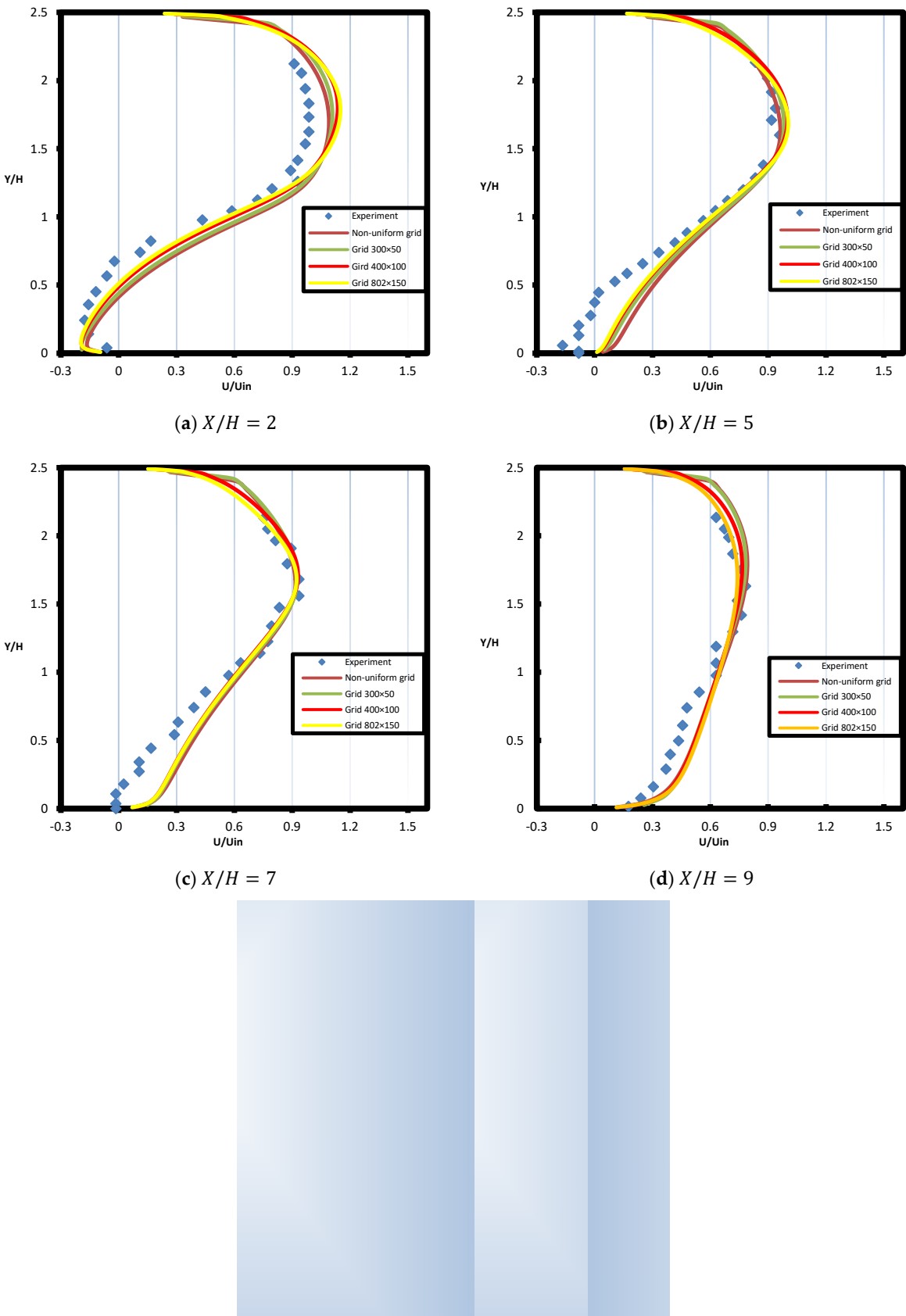

**Figure 3.** The mean air flow velocities for BFS duct flows with experimental data of [14] at various resolutions and various streamwise positions.

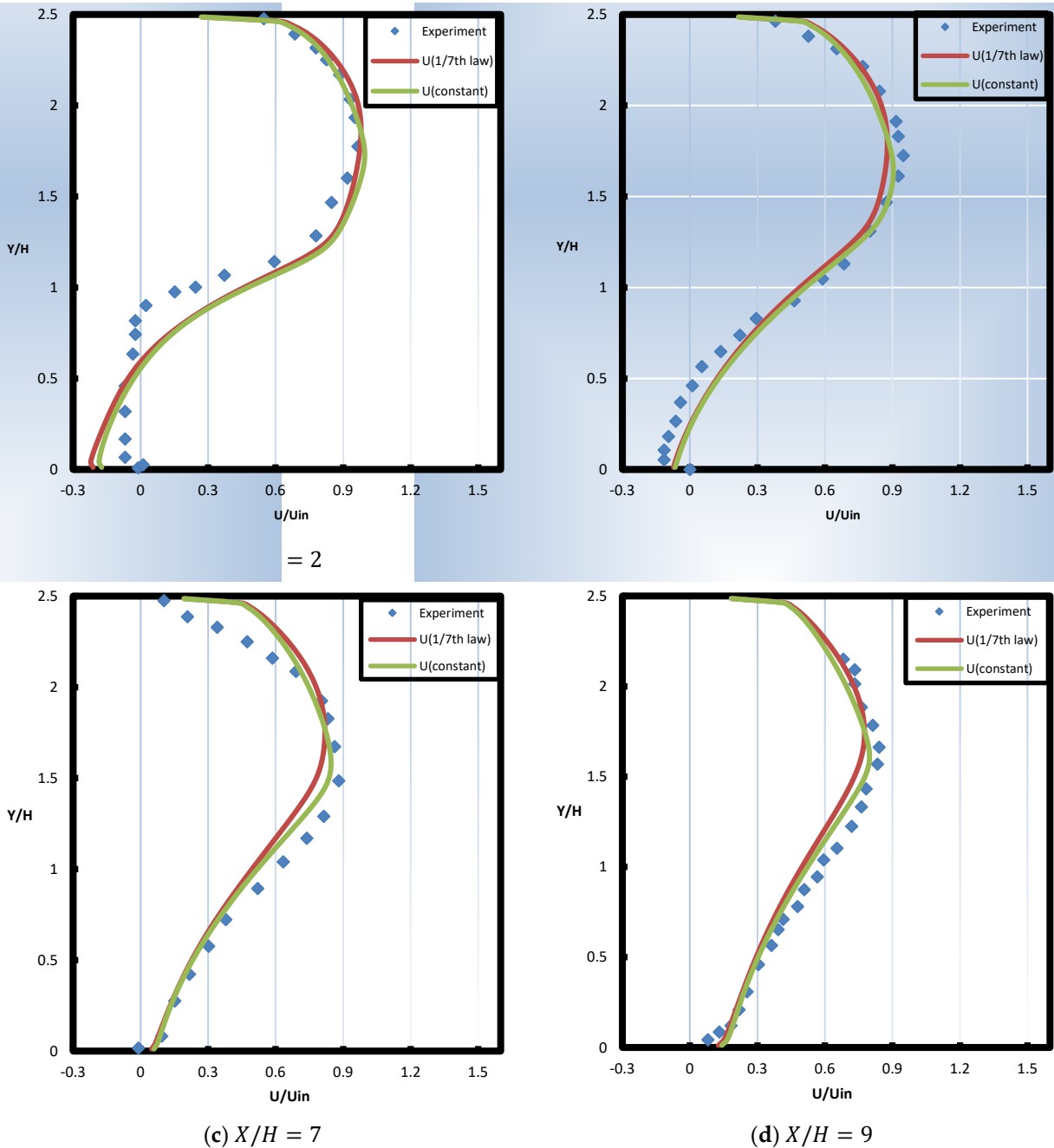

**Figure 4.** The mean air flow velocities for BFS duct flows with experimental data of [18,19] at different streamwise positions for different inlet conditions.

For each grid, the velocity profiles of the flow were calculated and illustrated in Figure 3, this figure demonstrates how well the model predictions and the experimental results are consistent. The results to the finest grids ($802 \times 150$ and $400 \times 100$) are also very close specialty near the straight wall. As a result, the standard k-$\varepsilon$ model used can effectively predict the back flow behavior behind the step wall.

Also a comparison with experiments for K.F. Yu et al. and and Bing W et al. [18,19] were validate the code. The geometrical parameters is upstream channel width ($h = 0.04$ m), step height ($H = 0.0267$ m), and length of down-stream ($L = 0.907$ m), The inlet gas velocity 9.1875 m/s.

After studing the grid meshing (not included in this paper), the grid resolution of $600 \times 100$ had been chosen. Figure 4, indicates the mean velocity with different inlet

conditions, 1/7th power law and constant velocity. The different inlet conditions had no effect on the results because the upper channel length used is sufficient to reach the fully developed flow.

Comparisons between predicted and measured normal Reynolds stresses are display in Figure 5. In addition, the Reynolds stresses in the mean direction $\overline{\tilde{u}\tilde{u}}$ is calculated using an anisotropic model proposed by [45]:

$$\overline{\tilde{u}\tilde{u}} = \frac{2}{3}k + \frac{(2c_{\tau 1} - c_{\tau 3})}{3} \frac{k^3}{\varepsilon^2} \left(\frac{\partial u}{\partial y}\right)^2 \tag{11}$$

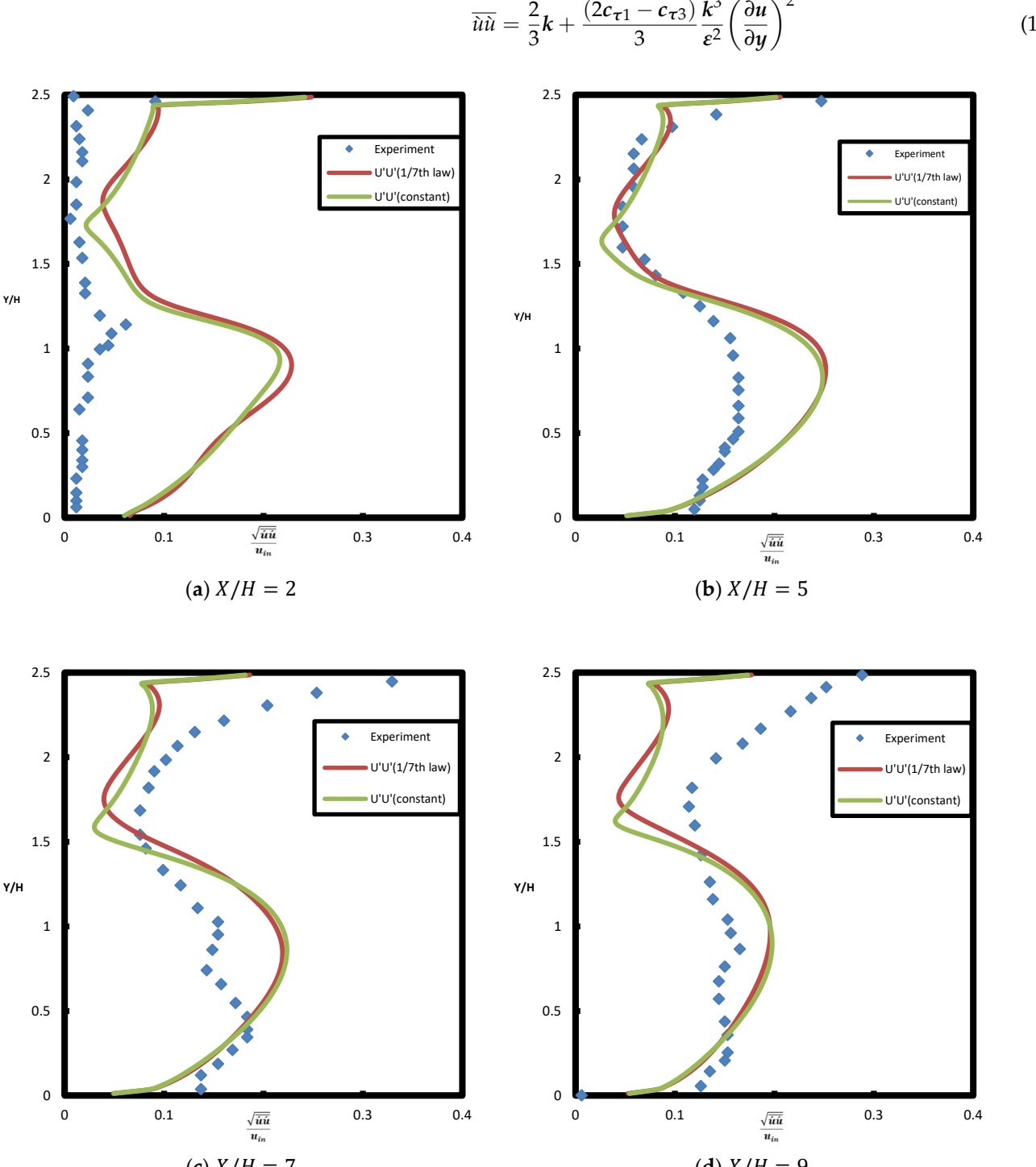

**Figure 5.** Comparison between computed square root of mean-mean fluid fluctuating velocity with experimental results of [18,19] at different streamwise positions for different inlet conditions.

The constants values are $c_{\tau 1}$ = 0.07 and $c_{\tau 3}$= 0.015 based on the optimised model parameters stated in [46,47], the model is unable to accurately anticipate the behaviour of Reynolds stress in the mean direction at $X/H = 2$. This is explained by the isotropic assumption included in the common standard k-$\varepsilon$ model. The latest locations profile at $X/H = 5, 7, 9$ are pretty well predicted by the model. In summary, the created model can accurately forecast the turbulent flow during the BFS, which motivates the authors to continue their theoretical research.

### 3.1.2. Validation of Axisymmetric Diffuser

A comparison with the experimental results of Singh and Azad [26] will be introduced below to validate the new technique called cell cutting, it works to get closer and reach a suitable solution. The diffuser's inlet radius ($R_{in}$ = 0.0508 m), down-stream pipe radius ($R_{out}$ = 0.101 m), and diffuser length ($L$ = 0.745 m) are all factors to be taken into account in this situation. The inlet velocity is 10.5 m/s. Figure 6, indicates the axial velocity in positions $X/H$ (0.23, 0.557, 0.7). ($H = R_{out} - R_{in}$), and $X$ is the location measured behind the step. We use the different resolution to illustrate the grid independence study with ($302 \times 68$, $602 \times 136$, $604 \times 227$).

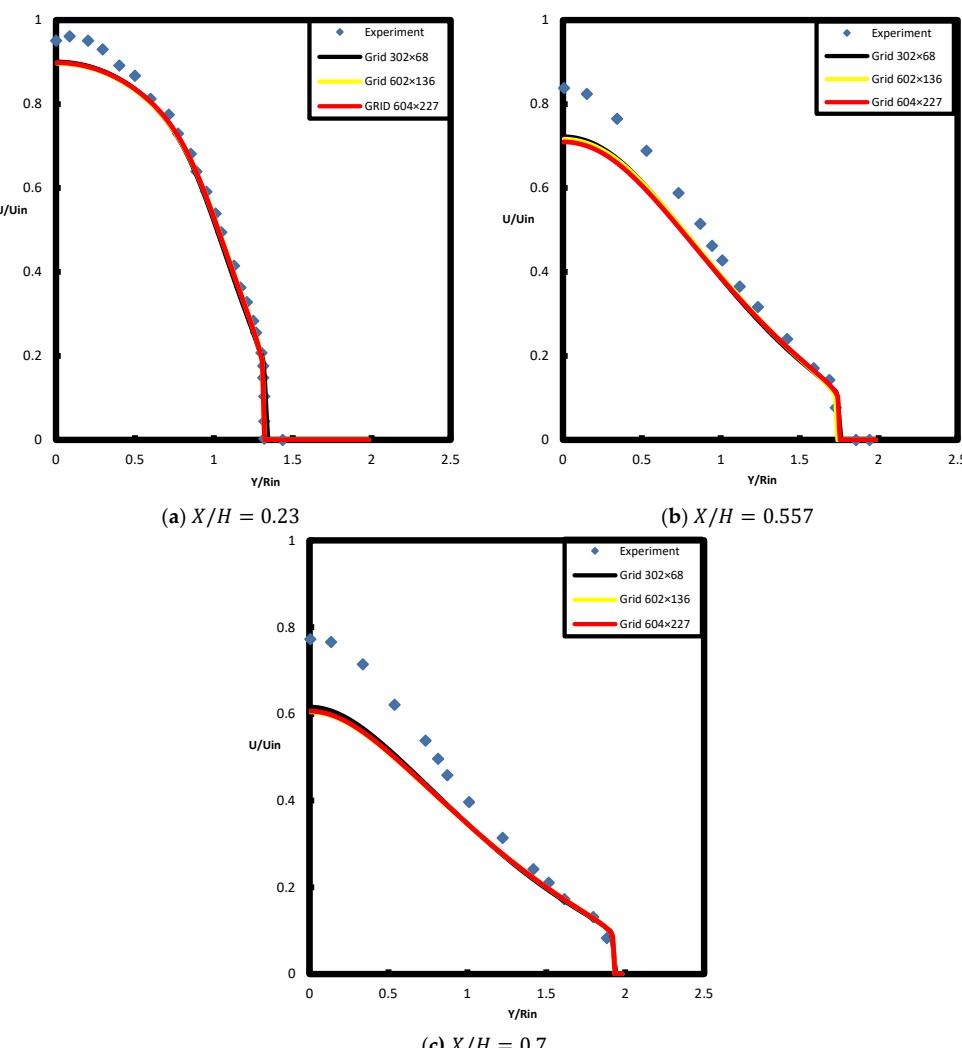

(**a**) $X/H = 0.23$ (**b**) $X/H = 0.557$

(**c**) $X/H = 0.7$

**Figure 6.** Comparison of axial velocity profiles at various positions through the diffuser using the current computational model and the experimental data found in [26].

Figure 7a, illustrates a comparison between the experimental data for the wall static pressure coefficient and the current numerical results. In particular, when applying the

conventional turbulence model, the comparison reveals good agreement between the current predictions and experimental results [26]. Just at diffuser entrance a steep decrease followed by gradually increase in the gradient is seen. Afterwards pressure decreases gradually in the down-stream pipe. With other valid to pressure, As reflect in Figure 7b, the change in pressure ($p = p_2 - p_1$) of diffuser expansion may be calculated simply by contrasting the variation between the upstream and down-stream totally formed pressure gradient lines extended to the step wall site ($x = 0$) with the experimental data in [44]. the wall static pressure coefficient defined as Equation (12):

$$C_p = \frac{p_{(x)} - p_{ref}}{\frac{1}{2}\rho U_{ref}^2} \tag{12}$$

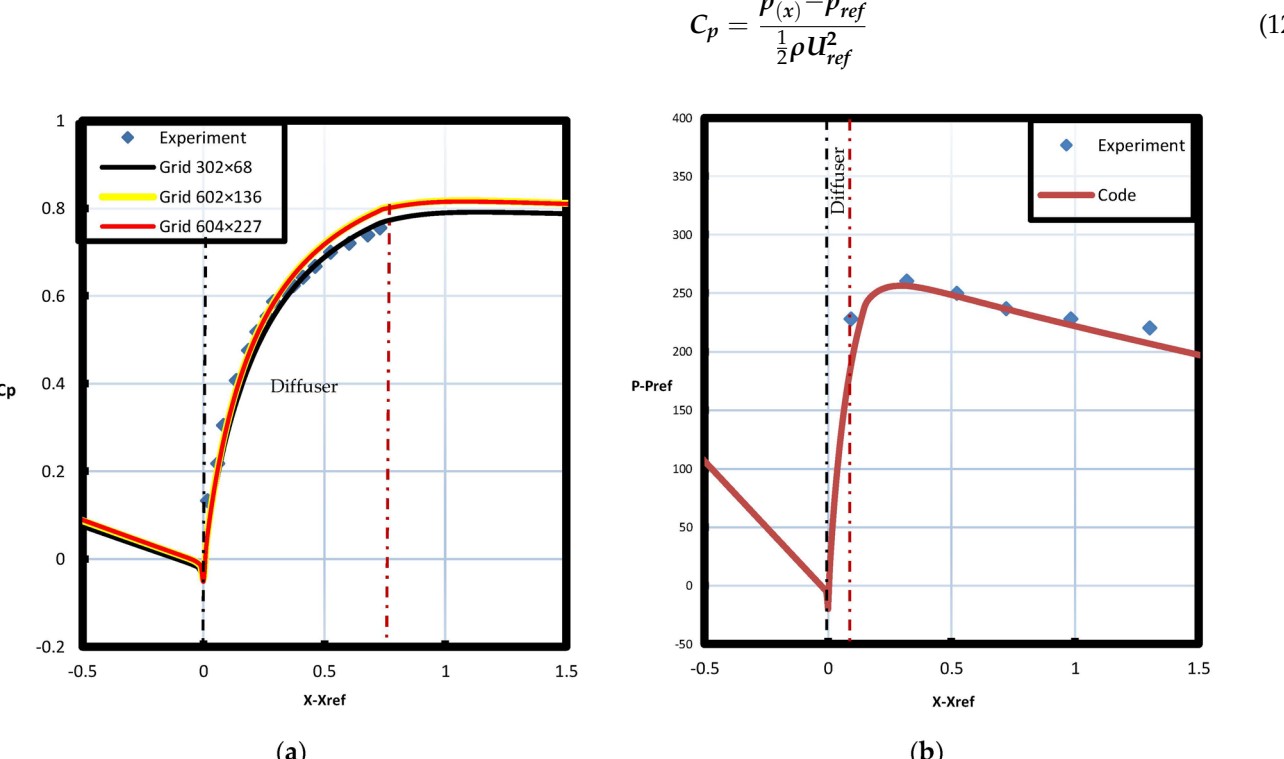

(**a**)　　　　　　　　　　　(**b**)

**Figure 7.** (**a**) Present numerical wall static pressure coefficient distribution with experimental data of [26], (**b**) Present numerical pressure distribution with experimental data of [44].

The local skin friction coefficient is shown in Figure 8, where $\tau_w(x)$ is the local wall shear stress at any longitudinal distance ($x$), $U_{in}$ is the inlet velocity, the local skin friction in Equation (13).

$$C_f(x) = \frac{\tau_w(x)}{\frac{1}{2}\rho U_{in}^2} \tag{13}$$

The friction coefficient approaches constant value in the fully developed part of the upstream pipe. This value is very close to analytical solution given by Blasuis equation for smooth wall in Equation (14), turbulent and fully developed pipe flow as in Schlichting [46,47]. The friction coefficient decreases through the diffuser due to adverse pressure gradient and the distortion of boundary layer.

$$Cf(x) = \ 0.079/Re_{in}^{0.25} \tag{14}$$

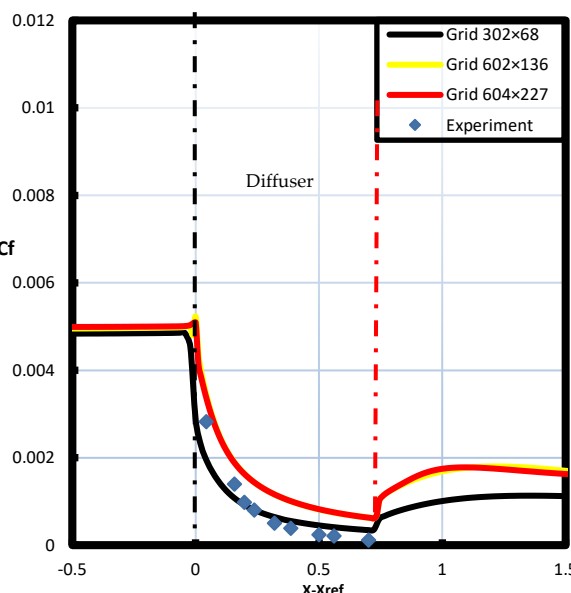

**Figure 8.** The current numerical and experimental data are compared [26] for streamwise distribution of local skin friction coefficient.

### 3.2. Effect of Area Ratio in Backwarad Facing Step (BFS)

One extremely major aspect in the practical field that has a big impact on fluid flow is the area ratios effect. This study investigates numerically the effect of the area ratios (AR) on turbulent flow, the effect of the velocity, streamlines, static pressure, local skin friction coefficient, and dimensionless wall distance y+ will present below. The geometrical parameters used in this section as the work of [18,19], by using different down-stream height to study the effect of AR, the down-stream height will be ($O = h + H$), shown in Table 2. All figures in this section use a grid size of ($600 \times 104$).

**Table 2.** Area ratios of backward-facing step.

| O | 0.08 | 0.1 | 0.12 | 0.14 |
|---|---|---|---|---|
| Area ratios (O/h) | 2 | 2.5 | 3 | 3.5 |

This research seeks to deepen our understanding of how (AR) influences the development of separation bubbles in a channel flow. According to results, recirculation zones have formed close to the step corners of the channel. The AR appears to have a considerable impact on the recirculation zones' size. The mean velocity reflects in Figure 9, with a different positions $X/H$ (0, 0.83, 2.506, 6, 77).

The comparison of backward facing step with constant inlet flow rate with different area ratios based on different outlet duct. The flow rate will be constant at every section of backward facing step, so the velocity times area should be constant ($\rho$ is constant) (incompressible), thus as area ratio increase the velocity should be decrease on the same flow rate by M. V. Otiigen [48].

Figure 9, presente that the velocities just after the step are very equal in maximum magnitude in all different area ratios, due to the equal kinetic energy of the inlet fluid flow to the step. After a suitable distance (X/O = 2.506), the effect of area change will start to appear, and the velocity will decrease as the area ratio increases.

The separation bubble is defined in two different ways. One is based on the dividing streamline, while the other is based on local skin friction [46,47]. As illustrate in Figure 10, the streamlines are displayed for various area ratios. These data demonstrate that as area ratios rise, the size of the separation bubble also rises. Figure 10, also indicates that the reverse flow is higher in the greatest area ratios, causing the separation bubbles increase.

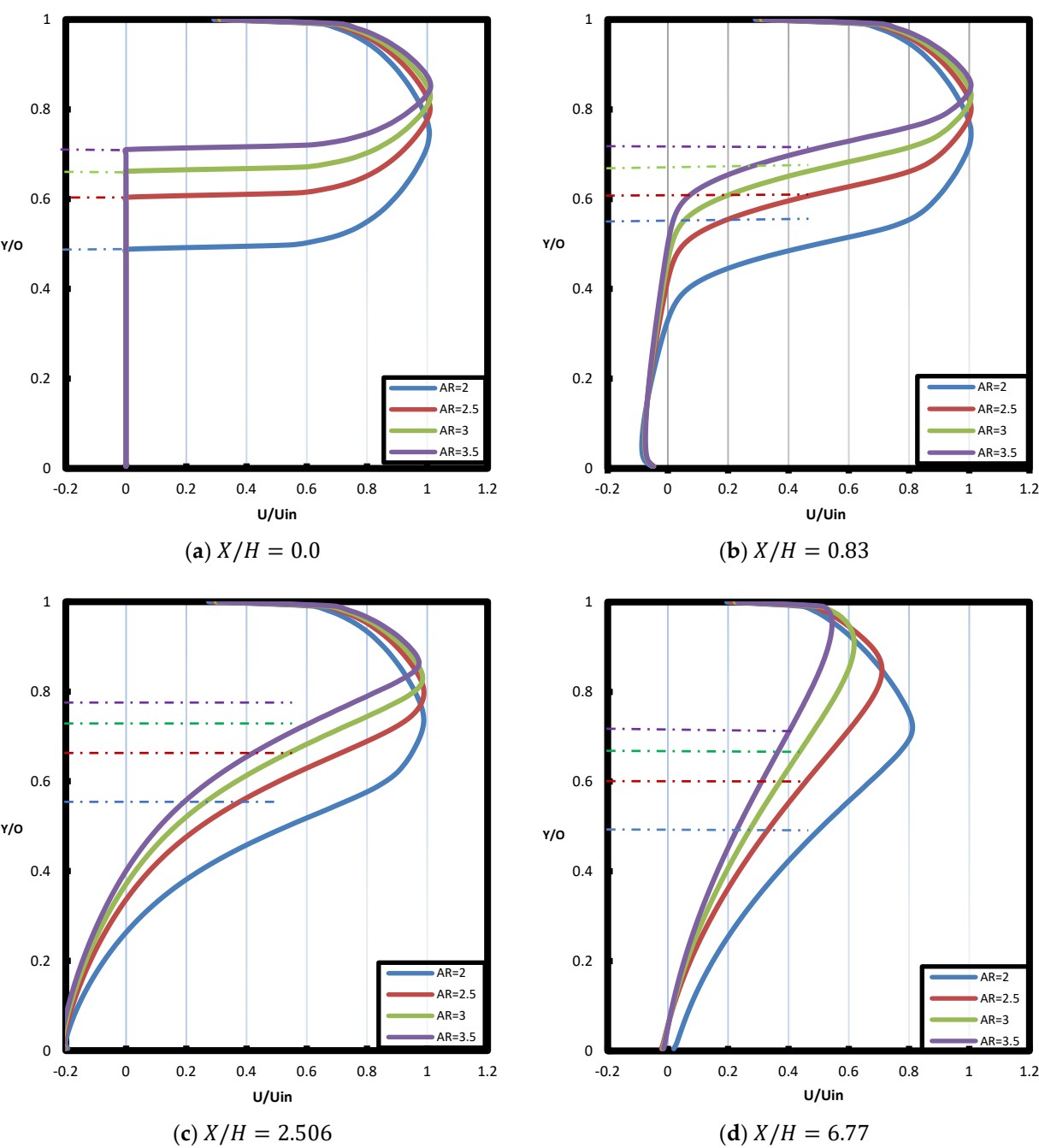

**Figure 9.** The mean air flow velocities for BFS duct flows with different area ratios and different position.

The longer separation bubble produced by a higher pressure gradient causes the reattachment length to generally grow as the area ratio does [24].

Figure 11, reflects the effect of the area ratios (2, 2.5, 3, and 3.5) on the static pressures coefficient of the two walls, there are no observed difference between the two graphs, This is related to the pressure change in the normal direction is neglected.

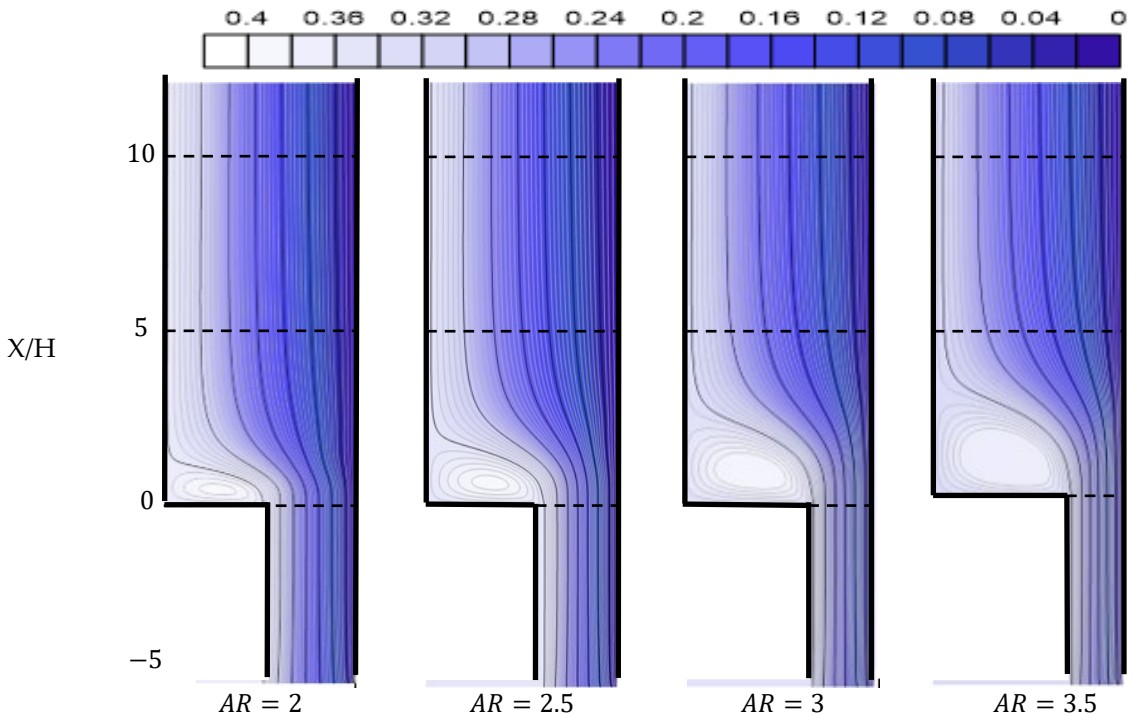

**Figure 10.** Air flow streamlines at various area ratios.

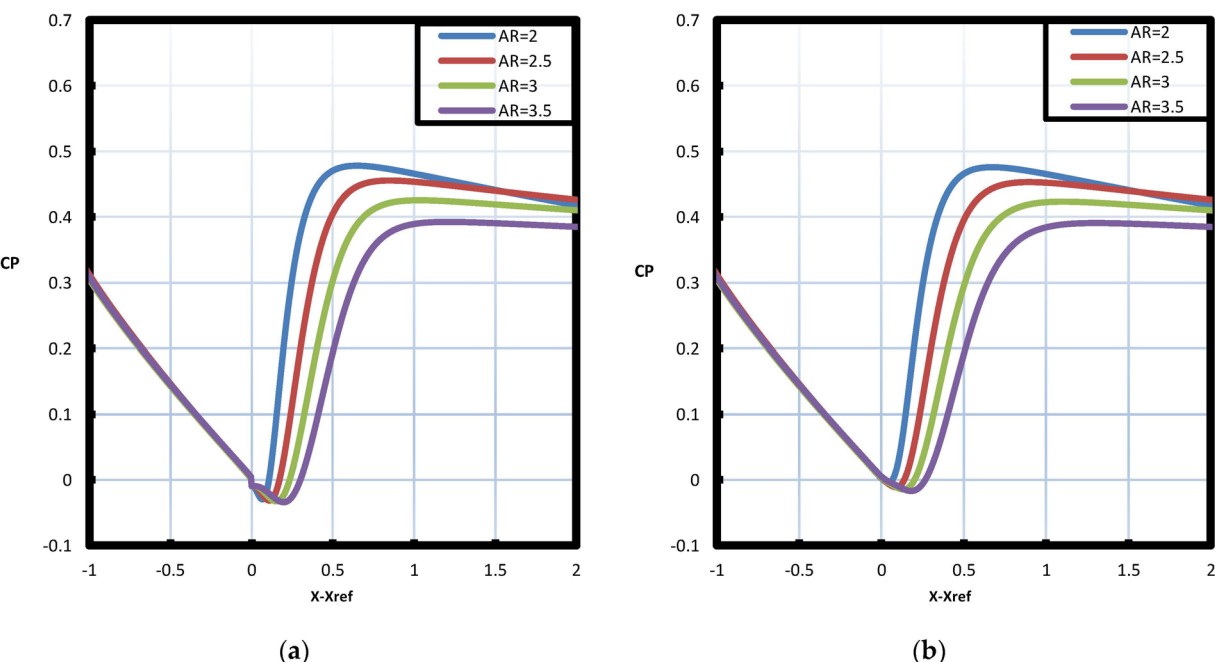

**Figure 11.** (**a**)The static pressure coefficient for BFS duct flows with different area ratios for upper wall, (**b**) The static pressure coefficient for BFS duct flows with different area ratios for lower wall.

The local skin friction coefficient reflects separation of the flow and the length of the reattachment is obovious in Figure 12a, which appear the local skin friction coefficient with a different area ratios. Figure 12b, displays the impact of area ratios on the dimensionless wall distance y+, all values of y+ is greater than 5 which leads to a good agreament with the standard k-ε models restrictions.

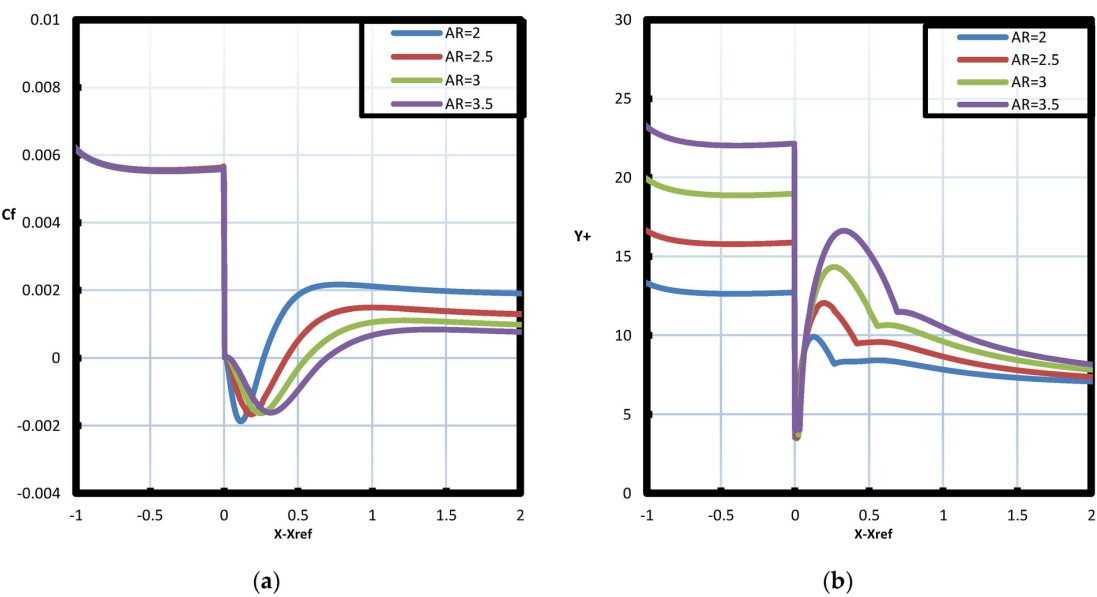

(**a**) (**b**)

**Figure 12.** (**a**) local skin friction coefficient with different area ratios, (**b**) dimensionless wall distance y+ with different arear ratios.

The static pressure coefficient depicts the effect of the area ratio; as the area ratio increases, the pressure decreases, this behavior is due to the seperation bubbles in the dowen-stream channel. The seperation bubble increases as shown in Figures 10 and 13, causing the pressure to decreases due to the streamline seperation from the wall. This results agrees with the experimental illustrated by M. V. Otiigen [48].

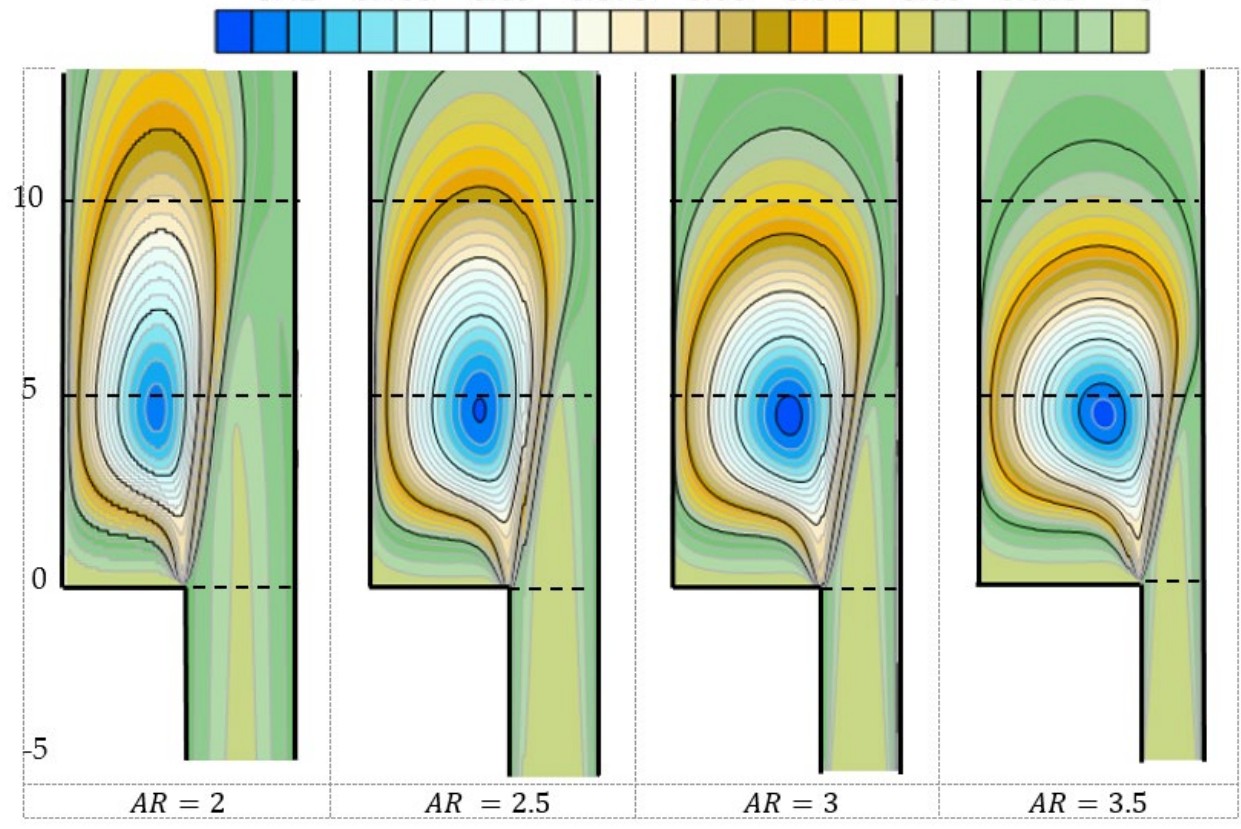

**Figure 13.** Area ratios' effects on the normalised turbulent kinetic energy $\frac{k}{0.5*u_{in}^2}$.

The turbulent kinetic energy has a relationship with the area ratios, when the area ratio increase the turbulence kinetic energy increase as display in Figure 13, so this means that as turbulent kinetic energy increases, the disturbances of fluid increase and the velocity decreases.

### 3.3. Effect of Area Ratios and Angles in Axisymmetric Diffuser
3.3.1. Effect of Angles in Axisymmetric Diffuser

The velocities, the static pressure coefficient, the local skin friction coefficient, the turbulent kinetic energy, and the streamlines will be identified. Several different angles will be studied, these angles 7°, 10°, and 14°. The instance being examined comprises the geometry of an abrupt expansion pipe with an upstream pipe radius ($R_{in} = 0.04$ m), the inlet gas velocity 9.1875 m/s.

Figure 14a depicts the difference between the angles (7°, 10°, and 14°) with same AR 2. The static pressure coefficient demonstrates the effect of the angles, as the angle increases, the pressure decreases this results agrees with the experimental illustrated by [24]. While Figure 14b reveals the local skin friction at different angles. The negative values of the skin friction coeffient appeared in the figure predictes the separation bubble of the flow, as the angle increases the seperation size increase. The effect of the seperation bubble is very obvious in Figure 14a, where the pressure increases as the same way in all angles in the difusser. The pressure decreases in the higher angle due to agreater seperation bubble generated. Figure 14c, indicate dimensionless wall distance y+.

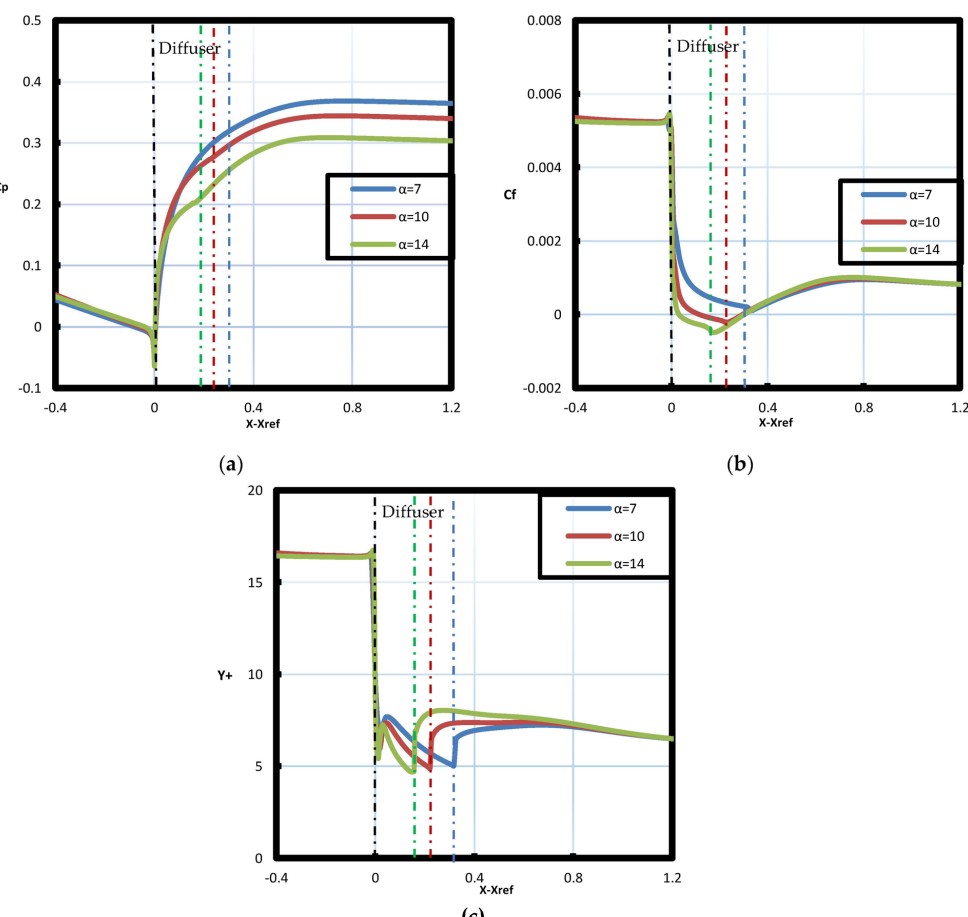

(a)

(b)

(c)

**Figure 14.** (**a**) The static pressure coefficient for axisymmetric diffuser with different angles with AR 2, (**b**) local skin friction coefficient with different angles with AR 2, (**c**) dimensionless wall distance y+ with different angles with AR 2.

The streamlines in the axisymmetric diffuser reflects the separation and eddies, allowing us to see the various angles that have an effect on creating the seperation, as presented previously in local skin friction help us to see the seperation region. when the angle increase, the seperation increase. Figure 15, displays the streamlines with different angles.

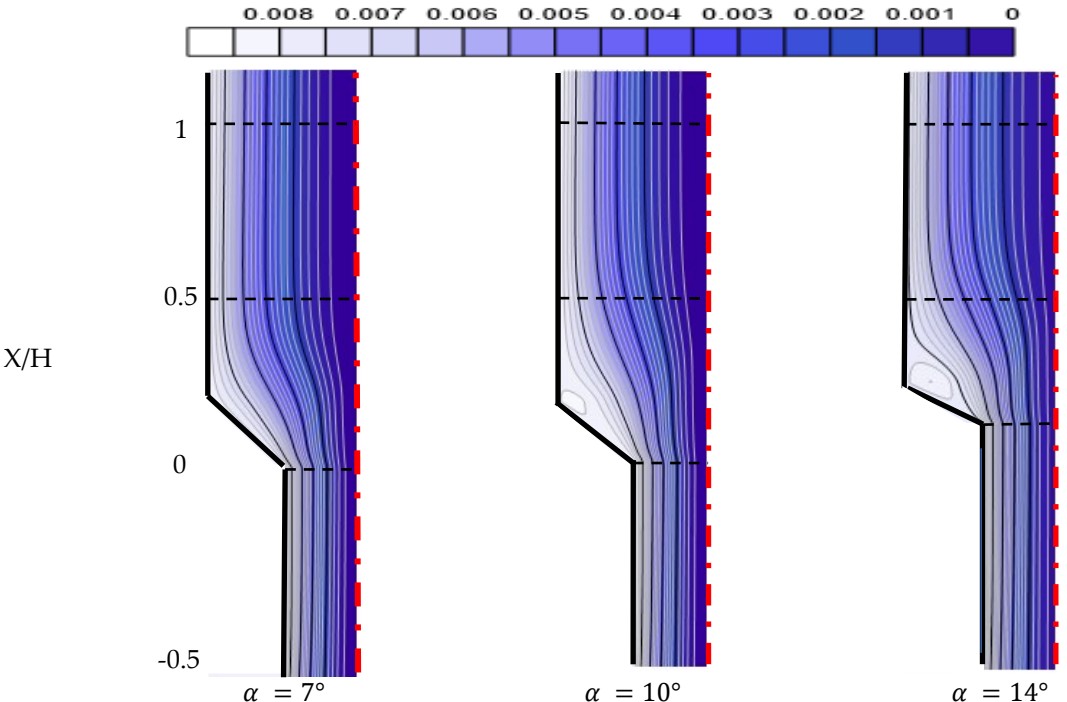

**Figure 15.** Streamlines of the air flow at different angles with a same area ratio is (2).

Turbulent kinetic energy measures the intensity of turbulence in a flow. The turbulent kinetic energy has a relationship with various angles. While angles grow, the turbulent kinetic energy increases, as illustrate in Figure 16.

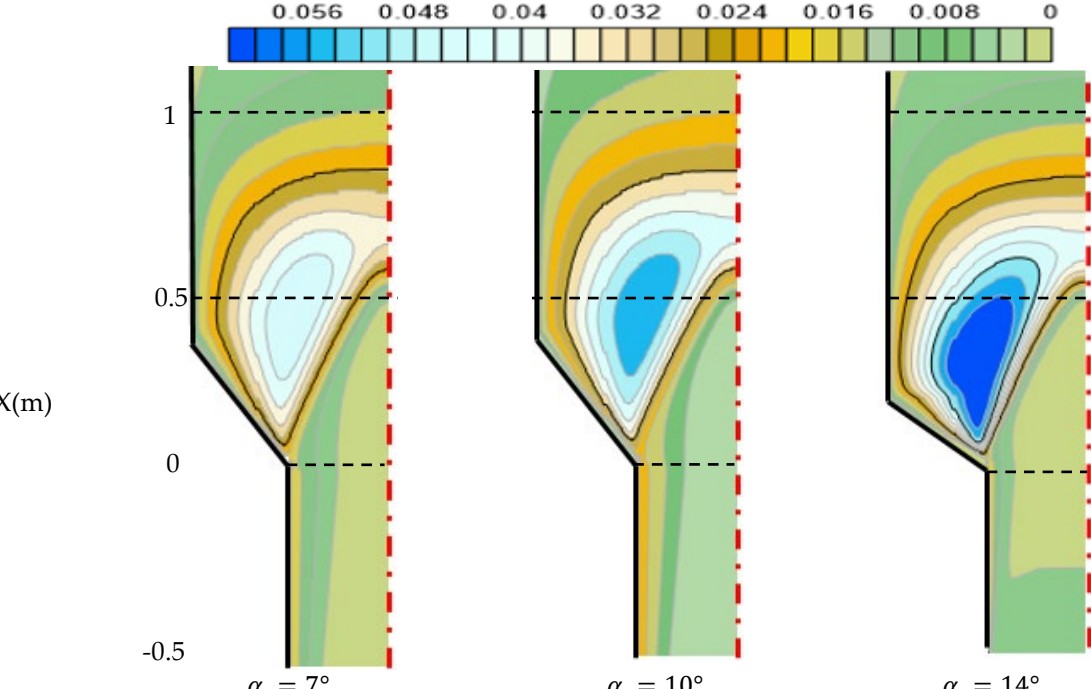

**Figure 16.** Effects of angles on normalized turbulent kinetic energy $\left(\frac{k}{0.5*u_{in}^2}\right)$ with area ratio (2).

### 3.3.2. Effect of Area Ratios in Axisymmetric Diffuser

A parametric study will be done on different area ratios to find out the effect that increasing the different area ratios can have. All figures in this section at a same angle of 10°. As the area ratios grow, the impact of static pressure, local skin friction, streamlines, and turbulent kinetic energy becomes more apparent. The local skin friction helps to find the sepration bubbles and reattatchment zones. Figure 17a, indicated the static pressure coefficient with a different area ratios. As the area ratio increses this will cause the pressure decresses, this behaviour is as the effect in backward-facing step (BFS) in Figure 11a. Figure 17b, demonstrate the local skin friction with a different area ratios, when the area ratios increase the separation size increase, and this behaviour same backward-facing step (BFS) in Figure 11b.

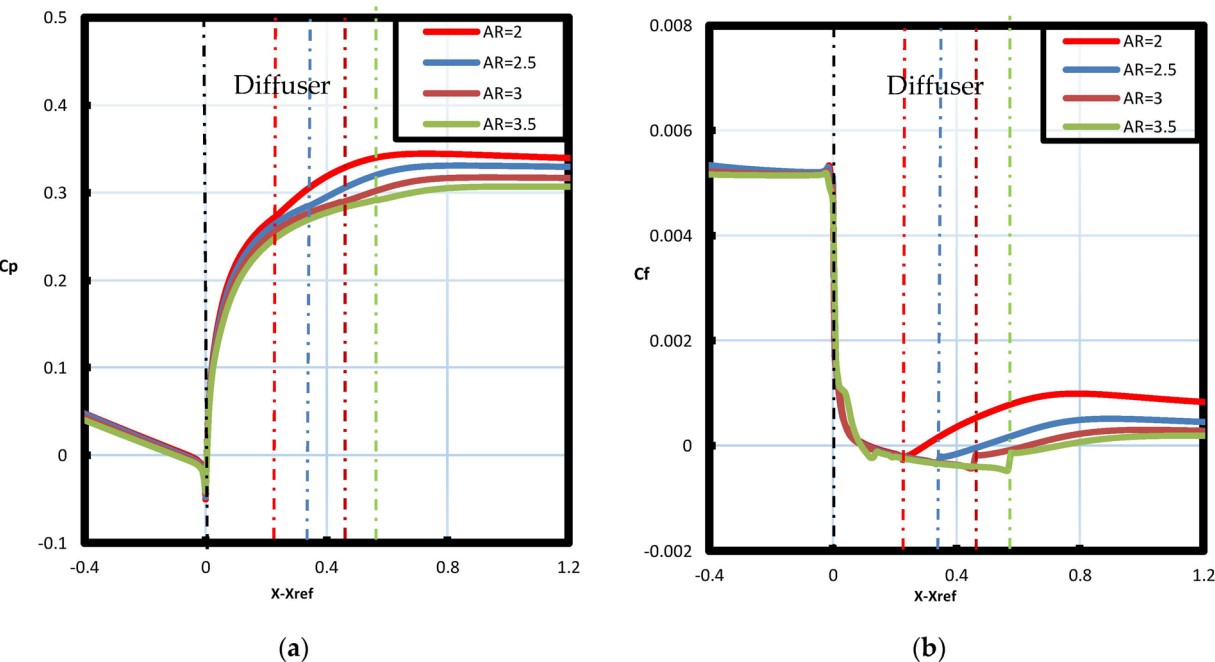

(**a**)  (**b**)

**Figure 17.** (**a**) The static pressure coefficient for axisymmetric diffuser with different area ratios with angle 10°, (**b**) local skin friction coefficient with different area ratios with angle 10°.

When fluids flow in parallel layers, there is no interruption or mixing of the layers, and at any given place, the velocity of each passing fluid stays constant throughout time. This kind of flow is known as streamline flow. A comprasion with a different angles with reespect area ratio and know the seperation region. Then the study will be a different area ratios with a same angle. The streamlines in the axisymmetric diffuser reflects the separation, to see the angle 10° with the different area ratios that have an effect on creating the eddies. Figure 18, appear the effect of area ratios to create the seperation with angle 10°. The separation size and eddies grow as the area ratios rise, therefore the streamlines assist in getting the data into agreement with the local skin friction.

While area ratios with same angle increase turbulent kinetic energy increase as show in Figure 19, the separation bubbles size is directly related to angles.

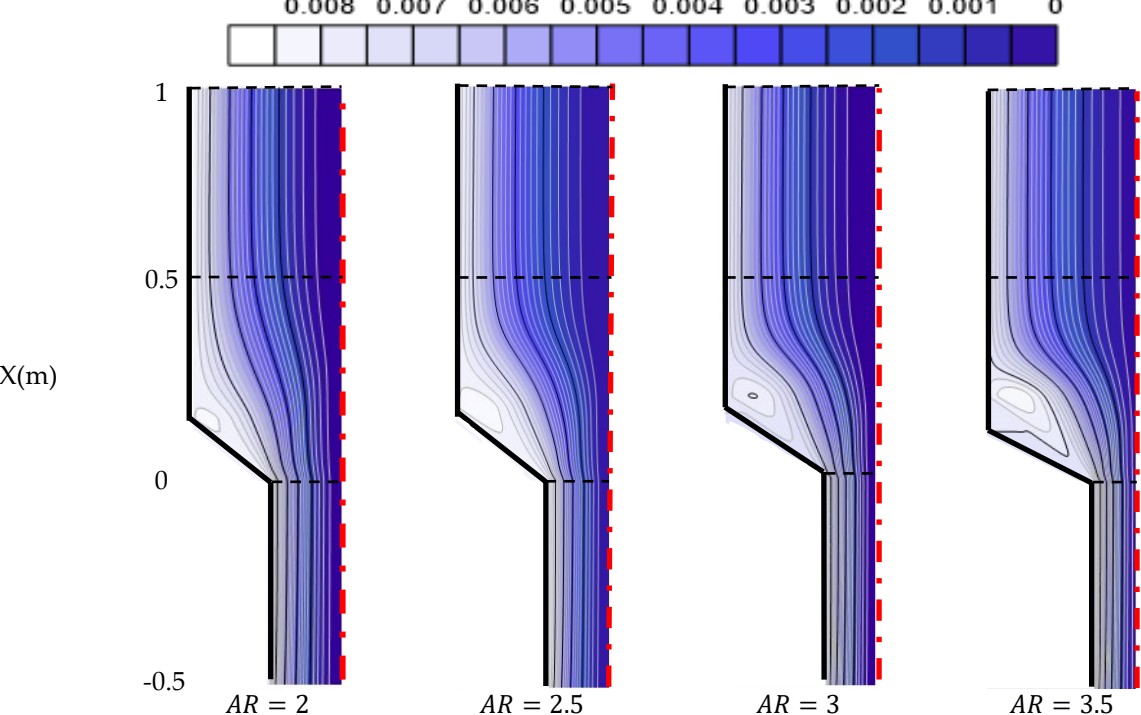

**Figure 18.** Streamlines of the air flow with a different area ratios and the angle is $10°$.

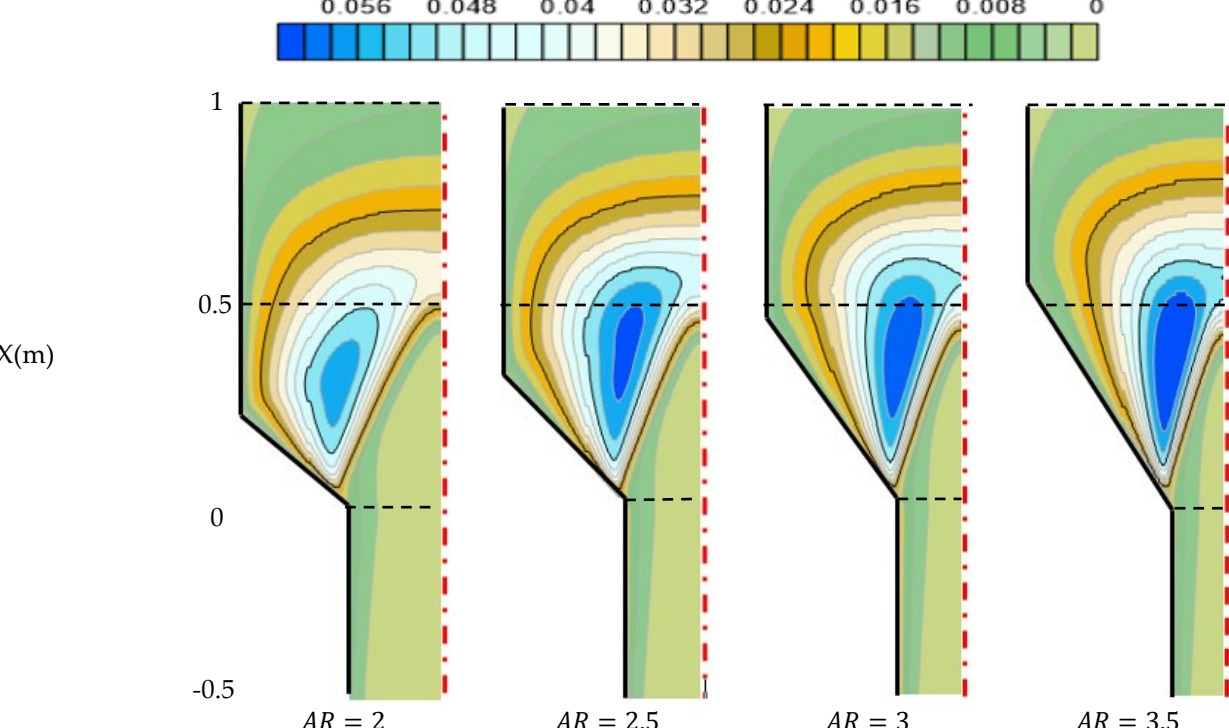

**Figure 19.** Effects of different area ratios on normalized turbulent kinetic energy $\left(\frac{k}{0.5*u_{in}^2}\right)$ with the angle $10°$.

## 4. Conclusions

The main purpose of this study is to analyze the differences between a grid study for the diffuser and how we can deal with the inclined wall in an axisymmetric diffuser.

This shows the cut-cell is very accurate because the numerical code is validated with experimental date. So the new technique reflects a good argument.

From the current investigation, the following primary conclusions may be drawn:

- The effect of area ratios in backward-facing steps (BFS) is that when the area ratios increase, the pressure decreases, the velocity decreases, and turbulent kinetic energy increase.
- The separation and eddies increase as the area ratios increase, so the streamlines reflect the impact of area ratios in reattachment.
- The cut cell for the axisymmetric diffuser that helps to get suitble numerical solution and get to be closer.
- When the angle is changed to increase while maintaining the same area ratio, the pressure decreases, the turbulent kinetic energy increase, and the eddies increase.
- When the same angle is used, the area ratios are changed to demonstrate the effect of area ratios with an axisymmetric diffuser. The pressure decreases, the turbulent kinetic energy increase, and the eddies increase

However, the performance of the backward-facing step (BFS) and axisymmetric diffuser for turbulent gas carrying solids will be taken into consideration in an extended study that will be conducted in the future, to enhance the performance of the computation, a nonlinear turbulence model may also be taken into account. The cut-cell technique works in small angles because the change of y-direction by change of x-direction equal $tan\theta$. However, this technique of cut-cell we use in small angles. The future work will be study to improve this limitation.

**Author Contributions:** Conceptualization, K.M.S., A.S.D. and R.M.A.; methodology, K.M.S., A.S.D. and R.M.A.; software, K.M.S. and A.S.D.; validation, K.M.S., A.S.D. and R.M.A.; formal analysis, K.M.S., A.S.D. and A.M.E.-R.; investigation, K.M.S., A.S.D. and A.M.E.-R.; resources, K.M.S., A.S.D., A.M.E.-R. and R.M.A.; data curation, K.M.S., A.S.D., A.M.E.-R. and R.M.A.; writing—original draft preparation, K.M.S. and A.S.D.; writing—review and editing, K.M.S., A.S.D., A.M.E.-R. and R.M.A.; visualization, K.M.S., A.S.D., A.M.E.-R. and R.M.A.; supervision, K.M.S., A.S.D., A.M.E.-R. and R.M.A.; project administration, K.M.S., A.S.D., A.M.E.-R. and R.M.A.; funding acquisition, K.M.S. and A.S.D. All authors have read and agreed to the published version of the manuscript.

**Funding:** This research received no external funding.

**Data Availability Statement:** Not applicable.

**Conflicts of Interest:** The authors declare no conflict of interest.

### Abbreviations:

The following abbreviations are used in this manuscript.

| | |
|---|---|
| BFS | Backward-facing step |
| RANS | Reynolds averaged Navier-Stokes equations |
| SIMPLE | Semi-Implicit Method for Pressure Linked Equations |
| LES | Large eddy simulation |
| DNS | Direct numerical solution |
| CFD | Computational fluid dynamics |
| FEM | Finite element method |
| FVM | Finite volume method |
| FDM | Finite difference method |
| RE | Reynolds number |
| AR | Area ratios |
| RSM | Reynolds stress model |

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
