# Peer review of "Numerical Investigation by Cut-Cell Approach for Turbulent Flow through an Expanded Wall Channel"

_axioms, doi:10.3390/axioms12050442_

Round 1
Reviewer 1 Report
The paper is very well written and the authors did a good job in ensuring the presentations are properly done. My only concern is that the topic is not new and I have some reservations to have it published under the title of Recent Advances in CFD and HT. Nonetheless the paper is written in such a way that any new postgraduate students will find it useful as a guide for them to start their work. For this reason, i will recommend it for publication.
Author Response
Thank you for your review and recommendation for publication. Thank you very much for your support. Thank you for the motivation you provide. The changes were marked with a red font. We share the concern about clarifying the contribution of this research.

Reviewer 2 Report
Summary of the Work
The aim of this study is to clarify the mechanisms governing turbulent flow in the axisymmetric diffuser by a cut cell technique and backward-facing step. The governing equations are the Reynolds averaged Navier-Stokes equations (RANS) with the standard k-ε model. In this investigation, the variables are the area ratios and the diffuser angles. The authors analyzed the effect of different area ratios (AR) on the flow's velocity, streamlines, and separation bubbles behind the step wall. Researching the angle difference impacting the diffuser, the authors also investigated a possible impact on the numerical code of the cell-cutting method. The main results obtained in this investigation (i.e., the effect of area ratio in the backward-facing step (BFS), the effect of area ratios and angles in an axisymmetric diffuser, etc.) are summarised in Section 4.
General Considerations
- Please ensure that all acronyms introduced in the manuscript are duly specified when they first appear in the text. For instance, in the Abstract please specify the acronym SIMPLE= Semi-Implicit Method for Pressure Linked Equations, etc.
- Please, check English; some typos were found.
- The limitations of the approach proposed by the authors are not discussed.
- There are some points that need to be clarified.
- The physical interpretations of the obtained results are missing (see suggestion 2) below).
The following suggestions are intended to help fill in some gaps in the manuscript.
Suggestions
1) The differential equations governing the turbulent flow are given by Eqs (1)-(5).
1a) Please, define variables \rho, U and V.
1b) As pointed out by the authors, to solve these equations, the boundary conditions must be specified. In sub-section 3.2 the authors mentioned very fleetingly that the "flow velocity is assumed to be a constant or a 1/7 power law profile and, with the wall function approximation, the slip boundary requirements are assumed for solid walls". For clarity, please write explicitly the expressions of the boundary conditions for variables \rho, U, and V.
1c) Eq. (7) establishes that \mu_eff is a sum of two contributions: \mu and \mu_t. Please define \mu and explain why, from the physical point of view, the turbulent contribution occurs in a simple additive form (and not, for example, in a multiplicative form or in a more complex manner).
2) Authors used the finite volume method with the Semi-Implicit Method for Pressure-Linked Equations (SIMPLE) algorithm for pressure velocity coupling. As known, the SIMPLE family of algorithms introduces pressure into the continuity equation. It is called semi- because the direct effect of pressure correction on velocity is dropped from the velocity correction equation, which causes very large pressure corrections. Although relaxations correct the pressure, these factors depend on the problems, which is a big minus for the SIMPLE algorithm. As a result, this algorithm destroys a good velocity field unless there is also a good pressure guess. Authors are asked to comment on this important aspect.
3) Figures (3)-(9) show the accuracy of the solutions found by authors' numerical code compared with various experiments (convergence of the solutions). However, if we consider the remark raised in the previous point, stability is also very important in the numerical solution of partial differential equations. So, what can the authors say about the stability of their solutions? For completeness, can authors also specify the order of accuracy of their numerical solutions?
4) In my opinion, another vulnerable aspect of this work is the lack of interpretation, from the physical point of view, of the results obtained. For instance,
4a) Please explain why if the area ratios in BFS increase, the pressure decreases, the velocity decreases, and the turbulent kinetic energy increases. Please explain why, from a physical point of view, keeping the ratio between the areas fixed, as the angle increases the pressure decreases, the turbulent kinetic energy increases, and the eddies increase.
4b) The authors showed that both turbulent kinetic energy and the eddies increase wherever the area ratios increase (for fixed angle), or when the angle increases (for fixed area ratio). This result is very important. Would the authors be able to interpret it in terms of Kolmogorov's theory of turbulence?
Conclusions
The research subject treated in this work is certainly topical as Backward Facing Step (BFS) is widely known for its application in the studies on turbulence in internal flows. As explained by the authors, the flow separation is caused due to the sudden changes in the geometry. This creates a zone of re-circulation and a point of flow reattachment. The work deserves attention even if there are some points that need further clarification. Authors are encouraged to take into account the suggestions above.
Author Response
Thank you for your review which encourage us to improve the manuscript. All of your comments have been addressed in the revised version and the changes were marked with red font. Below is point to point answers. We hope that you find the revised version of the manuscript suitable for publication.

Reviewer 3 Report
The authors studied the flow in field in both backward-facing step with different AR and axisymmetric diffuser of different angles. The work is well or and presented. But I think this topic has been studied by many researchers using both CFD and experiments. I think the novelty of this research is low.
And for the conclusion section, I do not see any insightful analysis result. Researchers with basic knowledge can provide these conclusions without even running the CFD simulation.
Author Response
Thank you for your review, which encourages us to improve the manuscript. All of your comments have been addressed in the revised version and we are looking forward to meeting your expectations. The changes were marked with a red font. We hope that you find the revised version of the manuscript suitable for publication.

Round 2
Reviewer 2 Report
The authors have satisfactorily answered all the questions raised in my previous report. In my opinion, this new version of the manuscript deserves to be published.
Author Response
Thank you for your review and recommendation for publication. Thank you very much for your support. Thank you for the motivation you provide.